# Newborn Screening for Fabry Disease: Current Status of Knowledge

**DOI:** 10.3390/ijns9020031

**Published:** 2023-06-05

**Authors:** Vincenza Gragnaniello, Alessandro P. Burlina, Anna Commone, Daniela Gueraldi, Andrea Puma, Elena Porcù, Maria Stornaiuolo, Chiara Cazzorla, Alberto B. Burlina

**Affiliations:** 1Division of Inherited Metabolic Diseases, Department of Diagnostic Services, University Hospital, 35128 Padua, Italy; vincenza.gragnaniello@aopd.veneto.it (V.G.); anna.commone@aopd.veneto.it (A.C.); daniela.gueraldi@aopd.veneto.it (D.G.); andrea.puma@aopd.veneto.it (A.P.); elena.porcu@aopd.veneto.it (E.P.); maria.stornaiuolo@aopd.veneto.it (M.S.); chiara.cazzorla@aopd.veneto.it (C.C.); 2Neurology Unit, St Bassiano Hospital, 36061 Bassano del Grappa, Italy

**Keywords:** Fabry disease, newborn screening, lysosomal storage disease, digital microfluidics, tandem mass spectrometry, second tier test, LysoGb3

## Abstract

Fabry disease is an X-linked progressive lysosomal disorder, due to α-galactosidase A deficiency. Patients with a classic phenotype usually present in childhood as a multisystemic disease. Patients presenting with the later onset subtypes have cardiac, renal and neurological involvements in adulthood. Unfortunately, the diagnosis is often delayed until the organ damage is already irreversibly severe, making specific treatments less efficacious. For this reason, in the last two decades, newborn screening has been implemented to allow early diagnosis and treatment. This became possible with the application of the standard enzymology fluorometric method to dried blood spots. Then, high-throughput multiplexable assays, such as digital microfluidics and tandem mass spectrometry, were developed. Recently DNA-based methods have been applied to newborn screening in some countries. Using these methods, several newborn screening pilot studies and programs have been implemented worldwide. However, several concerns persist, and newborn screening for Fabry disease is still not universally accepted. In particular, enzyme-based methods miss a relevant number of affected females. Moreover, ethical issues are due to the large number of infants with later onset forms or variants of uncertain significance. Long term follow-up of individuals detected by newborn screening will improve our knowledge about the natural history of the disease, the phenotype prediction and the patients’ management, allowing a better evaluation of risks and benefits of the newborn screening for Fabry disease.

## 1. Introduction

Fabry disease (FD, OMIM 301500) is an *X*-linked lysosomal disorder, caused by α-galactosidase A (α-GalA) deficiency, encoded by *GLA* gene, that leads to progressive accumulation of globotriaosylceramide (Gb3) and related glycosphingolipids (Figure 1) [1,2].

The clinical spectrum of FD is wide (Figure 2). Patients with a classic phenotype present with angiokeratomas, neuropathic pain, hypohidrosis, hearing loss and gastrointestinal symptoms. These symptoms can occur in early childhood before age 5 years, especially neuropathic pain and gastrointestinal symptoms [3]. In adulthood, the patients can show severe involvement of kidney, heart, central nervous (mainly cerebrovascular disease) and peripheral nervous system. Patients presenting with the later onset subtypes have cardiac, renal and neurological involvements, with a different degree of clinical severity, in adulthood [4,5]. The clinical manifestations in female heterozygotes also depend on the X-chromosome random inactivation that increases the phenotypic variability [6]. Unlike other X-linked disorders, females with Fabry disease often show clinical manifestations. One possible explanation, besides X-inactivation and skew deviation, is the ineffective cross-correction of the enzyme activity in vivo. Unaffected fibroblasts from Fabry heterozygotes mostly secrete the mature form of the enzyme, which lacks the high-uptake mannose-6-phosphate residues. This form cannot be efficiently endocytosed by the affected cells. Therefore, a less active enzyme can complement the activity of the cells lacking expression of the enzyme [7].

The diagnosis can be confirmed by the enzyme activity measurement in dried blood spot (DBS), leukocytes, plasma or fibroblasts in FD males, identification of glycosphingolipid accumulation (Gb3 and especially its deacylated form lysoGb3 in plasma, urine, tissues) and genetic analysis [8,9]. Several therapies are available: enzyme replacement therapy (ERT) with recombinant human-galactosidase A (alfa 0.2 mg/kg or beta 1 mg/kg), to be given intravenously biweekly [10,11], and oral pharmacologic chaperon (migalastat) in patients with amenable pathogenic variants [12].

FD is a pan-ethnic disease, but a particularly high incidence of the later onset form is reported in Taiwan, due to the high prevalence of the pathogenic variant c.640−801G>A (IVS4+919G>A) [13].

The therapy should be initiated as soon as possible on presentation of early signs [14,15]. However, diagnostic delay is common, due to the heterogeneous and non-specific symptoms that frequently arise when organ damage is already irreversibly severe [16,17,18]. In the absence of a detailed family history or in case of de novo variants, presymptomatic detection of FD can be achieved only through a newborn screening (NBS) program.

Here, we summarize the current state of newborn screening for FD, including our long-term experience. Finally, we give an overview of programs with some level of implementation worldwide, discuss these data and highlight advantages and limitations.

## 2. Methods

We searched PubMed and EMBASE until 28 February 2023, using the search terms “Fabry disease” and “newborn screening” or “Fabry disease” and “second tier test” or “newborn screening” and “lysoGb3”. The search was extended with synonyms for FD and matching terms or headings. We selected full-text articles in peer reviewed journals in the English language. References were cross-checked for additional relevant papers.

## 3. Results

### 3.1. Screening Methods

From a technology perspective, high-throughput newborn screening for FD may be feasible using various analytical approaches (Table 1). Among these, the most frequently used are digital microfluidics (DMF) and tandem mass spectrometry (MS/MS), because they are multiplexable with commercially available reagents.

Immune quantification applies microbead array technology with detection of fluorescence to determine the amount of each protein. It requires protein-specific antibodies that are currently not commercially available. Moreover, the method is not useful in cases where a non-functional protein is produced [19,20].

Fluorometric enzymatic assay uses a fluorogenic substrate (4-methylumbelliferyl-D-galactopyranoside). After overnight incubation, the fluorescence of the enzyme product 4-MU (4-methylumelliferone) is measured [21,22,23].

DMF enzymatic assay is a multiplex approach, also based on fluorometric enzyme activity assays. Digital microfluidics involves the transport of sub-microliter volumes of both sample and enzyme assay components over an array of electrodes under the influence of an electric field, by a process known as electrowetting [24,25,26]. The “spatial multiplexing” DMF method permits each LSD enzyme reactions to be performed in an individual droplet under its individually optimized conditions [27]. It is the fastest currently available method, enabling same-day result reporting [28].

MS/MS enzymatic assay uses an assay mixture containing the substrate and internal standard. After overnight incubation, and remotion of detergents, salts and excess substrate, the samples are introduced to a tandem mass spectrometer. The enzymatic reaction product is quantified by determination of the ion abundance ratio of product to internal standard for each sample. Since all products and internal standards have different masses, several enzymes could be analyzed together by MS/MS [29,30,31,32,33].

Comparison between MS/MS and fluorometric method: DMF and MS/MS allow for the determination of multiple enzyme activities on a 3 mm disc-punch. Both use reagent kits supplied by commercial vendors (Babies Inc and Perkin-Elmer Life Sciences, respectively) that are inexpensive and readily available, but MS/MS, in contrast to DMF, can be modified by any laboratory to include more enzyme assays or other markers in a single assay [34]. MS/MS assay needs overnight incubation and is performed in a 96-well format. DMF analyzes 40 samples within 4 h in 48-well cartridges. In an MS/MS assay, the pH is not optimized. For more enzymes that necessitate of different buffers, multiple incubations can be combined prior to a single injection into the mass spectrometer instrument [35], whereas with DMF, additional microfluidics cartridges and readers are required [32].

There are numerous reports claiming superior performance for MS/MS relative to fluorometry and DMF for LSD screening [31,36,37,38].

The superior analytical range of MS/MS compared to fluorometric assays has been described by several groups [31,36,39,40], providing a more accurate value of enzymatic activity, especially for very low values. This in turn predicts a lower number of screen positives. These data have been confirmed by retrospective comparative studies in Taiwan [41] and USA [36]. An additional advantage of MS/MS is that the substrates are closer in structure to the natural enzyme substrates, because incorporation of a fluorogenic group into the molecule is not required [36]. Comparison of false positive rates is complicated by the fact that the values depend on the chosen cutoffs and by the uncertainty in establishing a positive sample based on gene sequencing (given the large numbers of variant of uncertain significance VUS). Probably the most important metric is the measured ratio of mean enzymatic activity of random newborns to that of affected samples. This ratio for MS/MS is 5- to 23-fold higher than that for DMFs, and it is expected to lead to a lower false positive rate [31]. Precision studies carried out by the CDC (Centers for Disease Control and Prevention, Atlanta, GA, USA) in an identical setting and with identical samples show that MS/MS provides improved assay precision over DMF [32]. However, a prospective comparative effectiveness study on 89,508 deidentified residual newborn DBS performed in California demonstrated that MS/MS, DMF and immunocapture showed high sensitivity, but lack in specificity, with need for improvement [34].

### 3.2. Second Tier Test

Specificity of the screening test can be improved with second tier biochemical or molecular testing. This latter is questionable because of the disclosure of genotypes associated with VUS and unclassified variants. Additionally, most NBS laboratories do not have the expertise to provide such second-tier testing [28].

As a biomarker, lysoGb3 can be measured in DBS by liquid chromatography-MS/MS technology [42,43,44]. In patients with FD, the concentration of lysoGb3 has been shown to have diagnostic value, and it correlates with phenotype and severity of manifestations (high levels in patients with classic phenotype, especially male, mild-to-moderately elevated levels in individuals with later onset phenotype) and females [42,43]. Moreover, it is a non-invasive marker for monitoring the disease during follow-up and treatment [42]. There are a few reports in which lysoGb3 has been evaluated in the neonatal period [33,42,45,46,47]. Increased values are very suggestive of FD [46,47], but normal lysoGb3 cannot exclude the possibility of later onset FD. In follow-ups of positive newborns with predicted later onset forms, the levels of lysoGb3 gradually increase with age, which might suggest a progressive and insidious accumulation. Thus, it may allow non-invasive investigation of patients in the presymptomatic period [45], despite it being yet unclear whether there is a critical threshold that justifies initiation of therapy [48].

### 3.3. Genetic Screening

Two molecular high-throughput methods, high-resolution melting analysis and Sequenom iPlex (Agena iPlex), have been investigated (Table 2).

High resolution melting (HRM) analysis: Primer sets were designed to cover the seven exons and the Asian common intronic pathogenic variant, IVS4+919G>A, of the GLA gene. The assay starts with PCR amplification in the presence of an appropriate DNA binding dye, followed by the formation of heteroduplex molecules and a final melting and analysis step. Variants are identified through a change in melting curve position, shape or deviated melting curve shape. Possible concerns may be the low sensitivity to identify all Fabry variants, especially those located at exons 2 and 6 because their amplicons are greater than 300 bp. The sensitivity to variable concentrations of nucleic acids or salts necessitates experience in analyzing the study results, because many parameters need periodic adjustment. Finally, HRM is not reliable for detecting male individuals, as the assay procedure depends on the formation of the heteroduplex [49].

Agena iPlex: It is a MassARRAY^®^ genotyping platform that analyzes nucleotide variations by mass spectrometry (MALDI-TOF), using a distinguishing allele-specific primer to amplify the extension products. Advantages of the Agena iPLEX assay are that stringent DNA quality control of the samples is not required, the procedure is relatively easy to perform in less than one day and the results can be interpreted by simply trained physicians and medical technologists. The limitation is that it only can detect known variants that have been designed into the assay panel. It is suitable when hotspot variants and common variations are known in a well-studied population. For example, in Taiwan, ~98% of Fabry patients carry variants out of a pool of only 21 pathogenic variants. An Agena iPLEX platform was designed to detect these 21 pathogenic variants and is being used [50,51].

### 3.4. Newborn Screening for FD in the World

Several NBS pilot studies and programs for FD have been implemented worldwide in recent years. However, state-based NBS programs still vary across countries, based on the economic cost of screening, local expertise and interest, political decisions and patient/family advocacy. A summary of available data on the reported NBS programs for FD is present in Table 3.

We reported the results from more than 5 years of NBS for FD in northeastern Italy, based on the determination of α-GalA enzyme activity in DBS using a multiplex MS/MS assay. Since 2015, 173,342 newborns (89,485 males) were screened. A genetic variant in the GLA gene (1:7879 newborns, 1:4068 males) was confirmed in 22 males. Among them, 13 carried a known pathogenic later onset variant (1:6883 males), and 9 had VUS or benign variants. The most common pathogenic variant was the later onset variant p.Asn215Ser (three patients). All patients were asymptomatic at the last follow-up (mean age 3 years), and none were receiving specific treatment. We did not detect any heterozygotes among the 83,853 newborn females screened [45].

Asia: In Taiwan, FD NBS started in 2006 using the fluorometric assay and then MS/MS [17,23,41,59]. The prevalence of FD is very high in Taiwan. The IVS4+919G>A variant is the most common (82%, about 1 in 1600 males). This variant can activate an alternative splicing in intron 4, causing insertion of a 57-nucleotide sequence between exon 4 and 5 of the αGalA cDNA and subsequent premature termination after seven altered amino acid residues downstream from exon 4 [13]. This variant has been reported to be prevalently associated with cardiac involvement in FD, although a small portion of patients carrying this variant have clinical manifestations [67]. Due to the low number of common pathogenic variants, and the high false negative rate in females (especially carrying IVS4 variant), a DNA-based NBS has been implemented (see above) [51].

In Japan, enzyme-based pilot screening started in 2007 [60], whereas in China only recently it has been introduced [61].

USA: FD was proposed for inclusion in the Recommended Uniform Screening Panel (RUSP) in 2008. However, because of uncertainties about the NBS test’s sensitivity, the prevalence of later onset variants, the unknown effectiveness of treatment and possible immunological response and the lack of prospective NBS and treatment studies, the Advisory Committee on Heritable Disorders in Newborns and Children (ACHDNC) rejected the proposal [68]. However, local laws supported by NBS advocates and parents allowed the implementation of systemic screening for FD in several states. In 2013, Missouri became the first state to screen all newborns for multiple LSDs (including FD), using a DMF-based fluorescent assay. Within the first 6 months, 43,701 specimens were screened, and 15 newborns were reported to have a genetic diagnosis of FD (1:2913) [14]. In 2014, Illinois initiated a pilot screening program for five LSDs, including FD, using MS/MS, and it was followed by statewide screening in June 2015 [64]. Pilot studies and programs were then started in other states.

Europe: In Europe, the screening for LSDs is in its early stage. The first pilot study worldwide was conducted in 2003 in northern Italy using a fluorometric assay on 37,104 consecutive male newborns, and 12 of them were identified with FD (1 in 3100 males, of which 1 with a classic form) [22]. In Austria, a small pilot study was performed with almost 35,000 samples, using an MS/MS methodology, during which nine patients with FD were identified [54]. There was also a pilot screening study in Hungary on about 40,000 samples using MS/MS, and three cases of FD were confirmed [55]. A small pilot screening was also performed in Spain, with a high number of benign variants detected in confirmatory tests [52]. The most relevant number of screened newborns in Europe was reported by our group in 2021 (see above) [45].

These programs have demonstrated the feasibility of newborn screening for FD. It is difficult to make comparisons among studies because of the differences in screening techniques, the classification of later onset, benign variant and VUS, the cutoffs (usually more conservative at the start of the program), the numbers of screened newborns, the geographical/ethnical variation and the changes in the classification of variants over time as knowledge accumulates. While the positive predictive value (PPV, the fraction of test positives that are true positives) is the gold standard for evaluating medical tests, currently PPVs for NBS in FD cannot be used as a performance metric due to difficulty in the definition of true positives, because of the uncertainly in the onset of disease symptoms. In addition, the use of the false positive rate has the same problems. The only metric that can be reliably obtained is the ratio between the number of screen positives normalized to the number of newborns screened (the screen positive rate). This ratio can be used for prospective pilot studies, pilot studies with de-identified DBS and prospective NBS programs (PPV can only be obtained from live NBS programs). However, the rate of screen positives depends on the cutoff value chosen by each NBS laboratory [69].

Furthermore, disease incidence is only an estimate assuming that all “true positive” infants will develop symptoms. Moreover, most studies do not distinguish between male and female newborns, and these latter are lost to NBS. For example, a Spanish study reported a very high incidence of disease (1:394 births), but the number of screened newborns was relatively low (*n* = 14,600). Moreover, only 1 patient had a known pathogenic variant, while 25/37 carried benign variants [52]. Different genetic backgrounds can also explain differences in incidence between countries. For example, in Taiwan, high incidence of the later onset *GLA* splicing variant (IVS4+919G>A) was detected [17]. However, all studies showed that FD is surprisingly more prevalent than previously estimated (1:40,000 by Desnick et al. [5]), especially the later onset form, which may represent an important unrecognized genetic disease.

### 3.5. Recommendations for Management of Positive Neonates

In most programs, if the αGalA activity is below the cutoff, the sample is retested (in duplicate), and, if the average of the duplicate persists below the cutoff, a second spot is requested. If the activity of the second spot is still below the cutoff, the infant is referred to the Clinic Unit for confirmatory testing. Samples are generally collected at a definite time in the first week of life, in several programs a second sample is required for premature babies (e.g., <34 gestational weeks and/or weight < 2000 g) and for sick newborns (e.g., those receiving transfusion or parenteral nutrition) or for samples with low activities for several enzymes due to a suspected preanalytical error [14,33,45,52,54,65,70].

Notably, different laboratories have different methods of determining cutoff values. Some laboratories use fixed cutoff values, established after the analysis of a set of normal control specimens. However, enzyme activity shows seasonal variation, related to the environmental temperature and humidity. Therefore, it is optimal to change the cutoff value for each test batch (e.g., % daily mean activity, DMA) [61]. Another method to identify suspected positive newborns is the use of postanalytical tools, such as Collaborative Laboratory Integrate Reports CLIR [71]. This multivariate pattern recognition software compares each new case with disease and control profiles and determines a likelihood of disease score. It integrates all possible permutations of enzyme assay ratios (in multiplex assays) but also demographic information, such as age at specimen collection, birth weight and sex (that can impact white cell count and therefore enzyme activity) [72]. Sanders et al. demonstrated that CLIR tools markedly improves the performance of each NBS platform (false positive rates and PPVs) [34].

Before the initiation of the NBS, protocols for definitive diagnostic tests, genetic counseling, follow up and treatment should be defined. Wang et al. developed guidelines for the diagnostic confirmation and management of presymptomatic individuals with lysosomal diseases, but more recent guidelines are lacking [73].

They suggested that once the diagnosis of FD has been confirmed, baseline diagnostic studies should be obtained. The infant should be seen by the metabolic specialist at 6-month intervals and monitored for onset of Fabry symptoms. For the individuals who have atypical variants, the strategy for regular follow-up and therapeutic intervention should be different from those with the classic type [73].

Gragnaniello et al. suggested that patients carrying variants associated with later onset forms should be monitored every 12 months, with clinical, instrumental and biochemical assessments [45]. A suggested diagnostic and follow up algorithm for presymptomatic patients is presented in Table 4. However, further investigations are needed to find the optimal way for monitoring and treatment timing, especially for patients with unclassified, VUS and later onset variants. In part, the difficulty is due to the poor correlations of residual enzyme activity and genotype with the clinical phenotype. Long term follow-up programs will allow a better definition of natural history, management and response to therapies, providing answers to the many outstanding question.

An important aspect of the management of positive newborns is the family screening. The combination of a detailed pedigree analysis and cascade genetic testing of at-risk family members can increase the number of patients identified, improve early diagnosis and clarify the pathogenicity of novel *GLA* variants [74]. In several studies it is reported that none of the infants with FD identified by NBS had a positive family history of FD or relatives with symptoms suggestive of the disease [45,64]. However, when a family genetic screening was performed, all studied families had previously undiagnosed family members, symptomatic or not [23,45]. Germain and the International Fabry Family Screening Advisory Board reviewed the literature on the family screening. For 365 probands, reported in 82 publications, 1744 affected family members were identified, which is equivalent to an average of 4.8 additional affected family members per proband [74]. A similar number has also been reported in a US study [75]. Potential barriers to the implementation of family genetic testing in some countries include associated costs, low awareness of its importance and cultural and societal issues [74].

### 3.6. Benefits and Challenges of FD Newborn Screening

In 1968, Wilson and Jungner described 10 principles that should be met prior to introducing a screening program [76]. According to this algorithm, FD reaches a score of 8. Although the authors proposed a disease scoring ≥ 8.5 for consideration for NBS programs, it should be noted that most LSDs don’t reach this threshold [77,78]. Thus, the implementation of FD newborn screening is still controversial.

Advantages and disadvantages of the FD newborn screening are summarized in Table 5.

As discussed above, reliable and effective methods for screening on DBS are available. The disease is more prevalent than previously clinically estimated. Newborn screening allows the early diagnosis and treatment, leading to a better prognosis. Moreover, the diagnosis of a newborn can be followed by family counselling and screening.

Later onset forms: The high incidence of later onset forms has raised ethical issues. Detection in the newborn period may have a negative psychological impact on parents and carries the risk to create “vulnerable children” [79] or “patients in waiting” [80], labelled and overmedicated. Moreover, it increases the costs for diagnostic laboratory testing and follow-up visits. The long-term follow-up of these infants will be essential to understand the natural history of the disease (which includes the manifestations of the different phenotypes) and the impact of early treatment. However, an early diagnosis of the later onset forms may have several advantages. A significant number of patients currently remain mis- or undiagnosed for many years [18]. The implementation of NBS could avoid this “diagnostic odyssey”, allowing timely treatment and subsequently better outcome [64]. Moreover, their identification allows physicians to perform cascade genotyping in at risk family members and identify undiagnosed relatives [74].

VUS/benign variants: NBS revealed a high incidence of polymorphisms, e.g., p.Asp313Tyr in European population and p.Glu66Gln in Japan, that are considered to be non-pathogenic based on in vitro expression, lysoGb3 concentrations in plasma, prevalence in healthy alleles and clinical and histological features [81,82,83,84].

Screening for FD also reveals a high prevalence of individuals with VUS or novel not yet classified genetic variants [56]. To determine whether these variants were pathogenic or not, functional (e.g., in vitro analysis, in silico tools), biochemical (e.g., lysoGb3), pedigree analyses and especially clinical manifestations should be performed [59] and often many years are needed to correctly classify a variant. For example, in Caucasian newborns, the most frequent genetic alteration reported is p.Ala143Thr [45,54,64,85]. Although it has been reported in association with both classic [86] and later onset (renal and cardiac) FD [87], it has been recently suggested to be a benign variant. Study in COS cells demonstrated a high residual enzyme activity of 36% [22]. Reported individuals with this variant showed unspecific symptoms, but no increase of plasma Gb3 and LysoGb3 [88] and no storage in tissue biopsies [89,90]. The high incidence of this variant in gnomAD database (5.06 × 10^−4^) and in the screening programs supports its lack of pathogenicity [91]. Furthermore, the αGalA enzyme is localized within the lysosome, suggesting normal trafficking [92]. Therefore, the pathogenicity of this variant is still debatable [74,93,94].

Female newborns: Enzymatic tests are not reliable for screening females due to random X-inactivation in different tissues. False negative results with enzymatic assays are about 40% [95,96] (up to 80% in Taiwan, where the IVS4 variant is predominant) [97], whereas DNA-based methods appear to be more sensitive and reliable. In most countries, mutational heterogeneity hampers the use of molecular analysis for high-throughput screening. Nevertheless, genetic assays may be considered in the near future, specifically due to improved technologies.

NBS cannot accurately distinguish classic from later onset forms: The prediction of disease severity is difficult, because enzyme activity and genotype do not clearly correlate with the phenotype and there is a large number of private *GLA* variants [57]. Moreover, the influence of modifier genes or other genetic factors on phenotype severity may be confounding, since individuals with the same *GLA* variant may occasionally have variable clinical manifestations during disease progression [93]. For heterozygotes, lyonization makes presymptomatic prediction of phenotypic severity impossible [73]. The only test that seems to predict a classic form on NBS is lysoGb3 measurement (elevated) [47], but, as discussed above, it cannot accurately differentiate the different forms.

Ideally FD NBS program should include: (1) a combined enzymatic and genetic approach, to perform a complete screening of all patients (males and females), and the enzymatic and genetic approach would be complementary in supporting the difficult interpretation of genetic variants; (2) an improved biomarker to use as second tier test. At the moment we do not have an ideal disease-specific biomarker for Fabry disease. Plasma lysoGb3 has been established as a good diagnostic biomarker for Fabry disease [98]. Nevertheless, lysoGb3 is not highly sensitive and highly specific as lysoGb1 for NBS in Gaucher disease [99]. LysoGb3 correlates well with the classic form, male sex, but normal levels cannot rule out a later-onset form. Furthermore, most of the literature regarding lysoGb3 refers to measurements in adult Fabry patients, and we need more data on values during infancy; (3) a newborn screening program for FD should be associated with a long-term follow-up program. Indeed, only such a clinical follow-up could determine the impact of this early diagnosis in the real-life management.

However, despite these limitations, the opinion of FD patients about NBS is favorable. Several studies explored the opinion of FD patients (*n* = 88) on NBS for FD (and other later onset diseases). Most participants agree with NBS. They felt NBS could result in better current health, eliminate diagnostic odysseys, lead to more timely and efficacious treatment and lead to different life-decision, including lifestyle, financial and reproductive decisions [100,101,102]. A different opinion is reported by genetic healthcare providers. Indeed, Lisi et al. evaluated the opinion of 38 genetic healthcare providers: FD was viewed less favorable that other LSDs due to later age of onset (potential for medicalization, stigmatization and psychological burden) and ambiguity regarding prognosis [103].

## 4. Conclusions and Future Directions

The frequency and technical practicability make NBS for FD feasible and affordable to be extended to large population. However, several issues still need further study:The lack of a second-tier test suitable to cover all the forms of the disease and reduce the recall rate;No biochemical detection of heterozygous females;The clinical interpretation of unclassified variants and VUS;The impact of early diagnosis on patients with later onset forms.

Efforts to capture long term follow-up data, associated to functional characterization of the controversial variants, studies of biomarkers and modifier genes to a better phenotype prediction and patients’ management will be crucial to address important ethical issues. To conclude, both the benefits and risks of NBS merit further study, underscoring the need for long term follow up.

## Figures and Tables

**Figure 1 IJNS-09-00031-f001:**
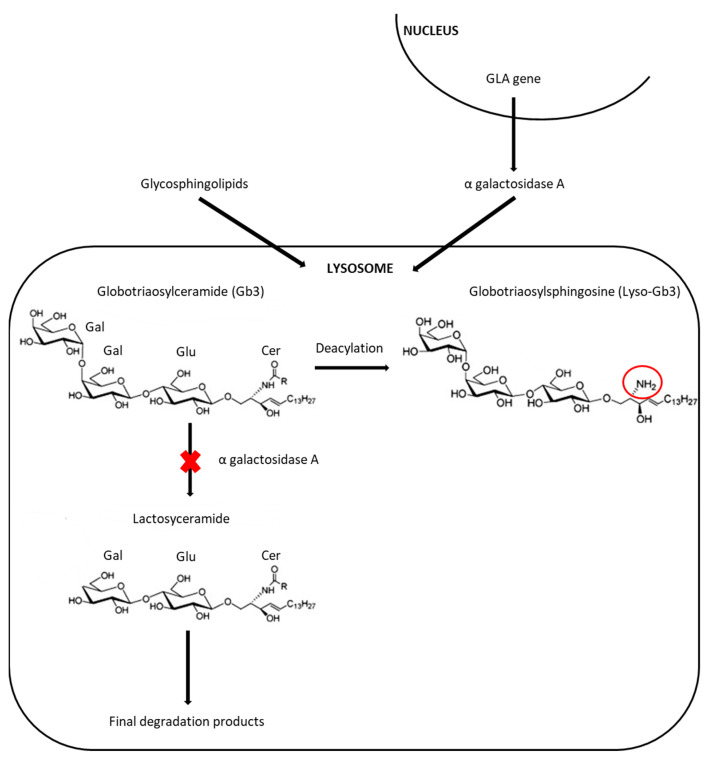
Metabolic pathway involved in FD. αGalA deficiency, due to the *GLA* gene pathogenic variant, leads to lysosomal storage of globotriaosylcramide (Gb3) and related glycosphingolipids (for example, its deacylated form, globotriaosylsphingosine or lysoGb3).

**Figure 2 IJNS-09-00031-f002:**
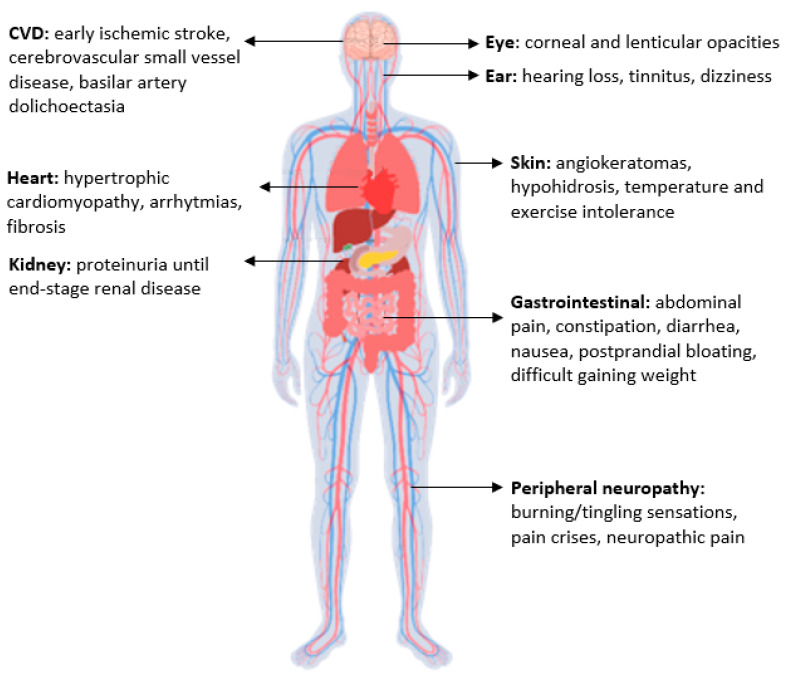
Principal signs and symptoms of FD. CVD: cerebrovascular disease.

**Table 1 IJNS-09-00031-t001:** Main characteristics of the available screening methods for FD newborn screening.

Characteristics	Fluorometry	Digital Microfluidics	Tandem Mass Spectrometry	Immune Quantification
Method	enzymatic assay	enzymatic assay	enzymatic assay	protein abundance
Multiplexable	no	yes	yes	yes
Incubation time	overnight	3 h	overnight	overnight
Assay conditions (specific pH, additives, buffers)	optimal	optimal	fixed pH (4.7)	not applicable
Interferences	low	low	very low	non-functioning enzyme
Analytical range	good	good	very good	not applicable
Instrumentation costs	low	low	high	low
Assay costs	low	intermediate	intermediate	low
Reagents	commercially available	commercially available	commercially available	not commercially available
Laboratory training	simple	simple	intermediate	Intermediate
Automation	Intermediate	high	high	Intermediate
Sample throughput	low	intermediate	high	low

**Table 2 IJNS-09-00031-t002:** Molecular assays for FD newborn screening.

Molecular Assay	Pros	Cons
High resolution melting	Cover the 7 exons and the IVS4 variant	Low sensitivity for variants located at exons 2 and 6 Sensitivity to variable concentrations of nucleic acid or salts Need of experience for periodic parameters adjustment Not reliable for males
Agena iPlex	Not stringent DNA quality control Easy, simple training Less than one day	Only known pathogenic variants

**Table 3 IJNS-09-00031-t003:** The most important FD pilot studies and screening programs worldwide.

Study Period	Country	Method	Type of Cutoff	Number of NBS Samples	Number of below Cutoff Samples	Number of below Cutoff Samples/100,000 Newborns	Confirmed Patients from Genetic Analysis *	Presumed Incidence **	Source of Data
**Europe**									
2003–2005	Italy	Fluorometric enzyme assay	fixed	37,104 (only males)	12 (m)	32 (m)	12 (m)	1:3100 (m)	Spada et al. [22]
2008	Spain	Fluorometric enzyme assay	fixed	14,600 (m 7575)	106 (m 68)	726 (m 898)	37 (m 20)	1:394 (m 1:378)	Colon et al. [52]
2010–2012	Italy	Fluorometric enzyme assay	fixed	3403 (m 1702)	0	0	0	/	Paciotti et al. [53]
2010 **	Austria	MS/MS	fixed	34,736 (deidentified)	28	81	9 (m 6)	1:3860	Mechtler et al. [54]
2011 ***	Hungary	MS/MS	fixed	40,024 (deidentified)	34	85	3	1:13,341	Wittmann et al. [55]
2015–2021	Italy	MS/MS	fixed	173,342 (m 89,485)	23 (m 22)	13 (m 25)	22 (m)	1:7879 (m 1:4068)	Gragnaniello et al. [45]
**Asia**									
2006–2008	Taiwan	Fluorometric enzyme assay	fixed	171,977 (m 90,288)	94 (m 91)	55 (m 53)	75 (m 73)	1:2293 (m 1:1237)	Hwu et al. [23]
2006–2018	Japan	Fluorometric enzyme assay	fixed	599,711	138	23	108 (m 64)	1:5552	Sawada et al. [56]
2007–2010	Japan	Fluorometric enzyme assay	fixed	21,170 (m 10,827)	7 (m 5)	33 (m 46)	6 (5 m)	1:3024 (m 1:2166)	Inoue et al. [57]
2007–2014	Japan	Fluorometric enzyme assay	fixed	2443	2 (m 2)	82	2 (m 2)	1:1222	Chinen et al. [58]
2008–2014	Taiwan	Fluorometric enzyme assay then MS/MS	fixed	792,247 (m 412,299)	764 (m 425)	96 (m 103)	324 (m 272)	1:2445 (m 1:1515)	Liao et al. [59]
2010–2013	Taiwan	MS/MS (compared with fluorometry)	fixed	191,767	79	41	64 (m 61)	1:2996	Liao et al. [41]
2015–2019	Taiwan	MS/MS	fixed	137,891	13	19	13	1:10,607	Chiang et al. [60], Chien et al. [46]
2019–2022	China	MS/MS	%DMA	38,945	21	54	3	1:12,982	Li et al. [61]
**USA**									
2011–2013 ***	California	MS/MS, immunocapture assay, DMF (comparative)		89,508 (m 44,664) (deidentified)	Variable based on method	Not applicable	50 (m 46)	1:1790 (m 1:1970)	Sanders et al. [34]
2013 **	Washington State	MS/MS	%DMA	108,905 (m 54,800) (deidentified)	16 (m 13)	15 (m 24)	7 (m 7)	1:15,558 (m 1:7800)	Scott et al. [62]
2013	Missouri	DMF	fixed	43,701	28	64	15 (m 15)	1:2913	Hopkins et al. [14]
2013–2019	New York	MS/MS	% DMA	65,605	31	47	7 (m 7)	1:9372	Wasserstein et al. [63]
2014–2016	Illinois	MS/MS	% DMA	219,793	107	49	32 (m 32)	1:6968	Burton et al. [64]
2016 ***	Washington State	MS/MS	% DMA	43,000 (deidentified)	8	19	6	1:7167	Elliot et al. [38]
**Latin America**									
2012–2016	Petroleos Mexicanos Health Services	MS/MS	fixed	20,018 (m 10,241)	5 (m 5)	25 (m 49)	5 (m 5)	1:4003 (m 1:2048)	Navarrete-Martinez et al. [65]
2017	Brazil	DMF	fixed	10,527	0	0	0	/	Camargo Neto et al. [66]

* We include all patients carrying a *GLA* variant. ** Disease incidence is only an estimate, assuming that all genetically confirmed newborns will develop symptoms. *** Because most pilot NBS are anonymous, confirmatory tests could not be performed. In these studies, samples that screen positive biochemically are genotyped. Abbreviations: m: males; DMA: daily mean activity; DMF: digital microfluidics; MS/MS: tandem mass spectrometry.

**Table 4 IJNS-09-00031-t004:** Confirmatory tests and follow-up of positive infants.

Timing	Suggested Tests
Diagnostic confirmation	Genetic analysis * (patient and parents), substrate quantification (plasma lysoGb3) and enzyme activity in leukocytes, lymphocytes or plasma (in males).
Baseline diagnostic studies	ECG, echocardiogram, ophthalmologic examination, renal function tests, plasma and/or urine GL3
Follow up every 6 months (classic form) or 12 months (later onset form)	Clinical examination (angiokeratomas, hypohidrosis, gastrointestinal symptoms, limb pain), kidney (eGFR according to Schwartz formula, microalbuminuria, proteinuria), cardiac assessments (ECG, echocardiography, 24-h holter), neurologic evaluation, plasma lyso-Gb3.

* Variants are classified according to published clinical reports and public databases.

**Table 5 IJNS-09-00031-t005:** Advantages and disadvantages of FD newborn screening.

Advantages	Disadvantages
Available methods for NBS on DBS	Enzyme based assays do not identify many female heterozygotes.
Approved treatments	Higher than expected numbers of later onset forms
Importance of early diagnosis and treatment, often delayed clinical diagnosis	Lack of definite guidelines for follow up and start therapy especially for later onset forms
Better knowledge of the natural history	Frequent detection of VUS or benign variants
Genetic counseling	Phenotype prediction can be difficult
Family screening	
High incidence, more frequent than previously expected	

Abbreviations: DBS: dried blood spot, NBS: newborn screening, VUS: variant of uncertain significance.

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
