# Peer review of "Newborn Screening for Fabry Disease: Current Status of Knowledge"

_2409-515X, 2023, doi:10.3390/ijns9020031_

Round 1

Reviewer 1 Report

The authors comprehensively review current knowledge concerning newborn screening for Fabry disease. There is in general no active comment on the information they collected. Therefore, I think the title in the system Newborn screening for Fabry disease: current status of knowledge is better than the topic on the manuscript Newborn screening for Fabry disease: is it really useful? A 2

review of the literature.

Because of the large amount of information, I can only comment on a few things that I am concerned.

1.     About the Taiwanese splice variant, the authors should also mention its standard nomenclature: GLA c.640-801G>A (IVS4+919G>A) at the first time

2.     The meaning of the IVS4+919 variant is, actually not clear. For example, only a small portion of males carrying this variant will have cardiac manifestations.

3.     Fabry disease itself is a late-onset disease, and there is no infantile type Fabry disease. We usually classify Fabry disease as classic or non-classic/cardiac variant.

Author Response

The authors comprehensively review current knowledge concerning newborn screening for Fabry disease. There is in general no active comment on the information they collected. Therefore, I think the title in the system “Newborn screening for Fabry disease: current status of knowledge” is better than the topic on the manuscript “Newborn screening for Fabry disease: is it really useful? A review of the literature”.

Because of the large amount of information, I can only comment on a few things that I am concerned.

  1. About the Taiwanese splice variant, the authors should also mention its standard nomenclature: GLA 640-801G>A (IVS4+919G>A) at the first time.

R: Thank you for the observation. We have included the cDNA variant c.640-801G>A in the text. 

  1. The meaning of the IVS4+919 variant is, actually not clear. For example, only a small portion of males carrying this variant will have cardiac manifestations.

R: Thank you for the evaluable comment. We have added that “a small portion of patients carrying this variant have clinical manifestations”.

  1. Fabry disease itself is a late-onset disease, and there is no “infantile” type Fabry disease. We usually classify Fabry disease as classic or non-classic/cardiac variant.

R: Thank you for your important comment. We have changed “infantile” with “classic” in the manuscript.

Reviewer 2 Report

The authors described the Newborn screening for Fabry disease: current status of knowledge . This review article is valuable and understandable.

I like this review. In the CHALLENGES OF FD NEWBORN SCREENING secsion,

I would like to request the description or discassion of ideal FD NBS system which should be realized in the future.

Author Response

The authors described the Newborn screening for Fabry disease: current status of knowledge . This review article is valuable and understandable.

I like this review. In the CHALLENGES OF FD NEWBORN SCREENING section,

I would like to request the description or discussion of ideal FD NBS system which should be realized in the future.

R: Thank you for your comment. Your request is crucial and very challenging. We summarized in the text as follows: “Ideally FD NBS program should include: 1) a combined approach enzymatic and genetic, to perform a complete screening of all patients (males and females); enzymatic and genetic approach would be complementary in supporting the difficult interpretation of genetic variants; 2) an improved biomarker to use as second tier test. At the moment we do not have an ideal disease-specific biomarker for Fabry disease. Plasma lysoGb3 has been established as a good diagnostic biomarker for Fabry disease (Burlina et al, 2023). Nevertheless, lysoGb3 is not highly sensitive and highly specific as lysoGb1 for NBS in Gaucher disease (Revel-Vilk e al, 2020). LysoGb3 correlates well with the classic form, male sex, but normal levels cannot rule out a later-onset form. Furthermore, most of the literature regarding lysoGb3 refers to measurements in adult Fabry patients, and we need more data on values during infancy; 3) a newborn screening program for FD should be associated with a long-term follow-up program. Indeed, only such a clinical follow-up could determine the impact of this early diagnosis in the real-life management.”  

Reviewer 3 Report

This paper is a timely literature review of the published information available on newborn screening for Fabry disease.  I think it will be useful as a reference as other nbs programs consider adding Fabry disease to their panels.  However, to make it as effective as possible, I would suggest a thorough editing of the paper with an eye to run-on sentences and organization.  Also make sure that there are transitions that make sense between topics Start with the abstract. 

Terminology note: 

Mutation vs variant:  Per guidelines, the preferred terminology for mutations is "pathogenic variant" if the change is disease causing, consider updating throughout the paper an in figure legends.

Late-onset vs later-onset vs. non-classic:  Each of these terms in used to describe the population of patient who do not have classic Fabry disease.  It seems to be preferred to use non-classic or later-onset; however all three terms are used in the paper.  Chose one and be consistent.

Specifics below:

Abstract:

lines 12-14, 16-18, and 20-22 each have several long sentences that needs to separated into shorter sentences each focused on one idea for clarity. 

lines 20-24, join together concepts that should be separated out such as methods being developed and then addressing limitations of NBS such as enzyme testing on females to improve idea flow.

Paper:

line 48-52, consider providing more detail on "childhood" vs. "youth" as there is stronger data available based on age categories.  "Youth" is too unclear a category to make sense in this context.   You have already referenced the Hopkin et al paper that can address much of that timing.  Consider Laney et al Genet Med, 17(5), 323-330. doi:10.1038/gim.2014.120  to better address childhood onset of symptoms to support NBS and need for early diagnosis.

In this section also consider reworking the separation out of females in the natural history discussion.   As females under age 5 with classic FD do have symptoms and in later onset do have cardiac/renal issues, consider just referring to classic and later onset FD without qualifying males/females and then the final statement can comment on phenotypic variability in females.

Also, the phenotype in women goes beyond X-inactivation, please add some nuance to that piece of natural history too.  (line 52, line 61, and other locations) See articles by Michael Beck  Beck M, Cox TM. Comment: Why are females with Fabry disease affected? Mol Genet Metab Rep. 2019 Oct 22;21:100529. doi: 10.1016/j.ymgmr.2019.100529. PMID: 31687338; PMCID: PMC6819736.and Esther Maier https://onlinelibrary.wiley.com/doi/10.1111/j.1651-2227.2006.tb02386.x

Figure 2:  consider adding depression/anxiety too CNS manifestations

Line 62, remove ( ) comment about X-inactivation as it doesn't add to discussion here and is oversimplified

In results section/screening methods:  consider beginning the section with lines 89 explaining why different approaches are more frequently used in the beginning to set the stage for the comparison beginning line 115.  Otherwise it is not clear why those 2 are being pulled out for comparision.

Table 2:  rework table to provide consistent data in results section for easier visual comparison.  Also comment on if the PPV includes VUSs or variants.

Section 3.3.  It seems important to handle this DNA-based methods as a section and remove the direct correlation to females.  The challenges to diagnosing females are very important but the paper would be clearer if the issues was separated from the DNA-based methods.  Perhaps a separate section.

line 206, consider replacing "parent" advocacy with "patient/family advocacy" to better reflect issues

Table 4- Given size of table, format to fit on one page or repeat headers to it can be effectively read.  Also comment on confirmed patients from genetic analysis if it includes VUS/A143T and perhaps number of classic/later onset in confirmed column.  The incidence should be classic/later onset as well.

Have you considered combining data from same NBS testing programs if the publications are on the same population/methodology?

Section 3.6.  Consider a short paragraph about the advantages of NBS for FD line 352 before jumping into challenges.  Have you considered information on the early onset of symptom as an addition to support for NBS in FD?

Line 382.  I'm not sure this conclusion is fully supported yet.

Table 6: disadvantages-  Diagnostic testing can distinguish infantile from non-classic in many cases.   Consider modifying this cell.

Overall the quality of English terms is good.  There are few interesting choices such as line 319 "instrumental", but over all well done.  However, there needs to be editing for English grammar.  I would suggest a thorough editing of the paper with an eye to run-on sentences and organization. 

Author Response

This paper is a timely literature review of the published information available on newborn screening for Fabry disease.  I think it will be useful as a reference as other nbs programs consider adding Fabry disease to their panels.  However, to make it as effective as possible, I would suggest a thorough editing of the paper with an eye to run-on sentences and organization.  Also make sure that there are transitions that make sense between topics Start with the abstract.

Terminology note:

Mutation vs variant:  Per guidelines, the preferred terminology for mutations is "pathogenic variant" if the change is disease causing, consider updating throughout the paper an in figure legends.

R: Thank you for the suggestion. We have modified the terminology in the manuscript.

Late-onset vs later-onset vs. non-classic:  Each of these terms in used to describe the population of patient who do not have classic Fabry disease.  It seems to be preferred to use non-classic or later-onset; however all three terms are used in the paper.  Chose one and be consistent.

 R: Thank you for your evaluable comment. We have modified with “later-onset” in the manuscript.

Specifics below:

Abstract:

lines 12-14, 16-18, and 20-22 each have several long sentences that needs to separated into shorter sentences each focused on one idea for clarity.

R: Thank you for your important comment. We have shortened the sentences as follow:

- lines 12-14 “Patients with a classic phenotype usually present in childhood as a multisystemic disease. Patients presenting with the later onset subtypes have cardiac, renal and neuro-logical involvements in adulthood. Unfortunately, the diagnosis is often delayed until the organ damage is already irreversibly severe, making specific treatments less efficacious.”

-lines 16-18 “This became possible with the application of the standard enzymology fluorometric method to dried blood spots. Then, high-throughput multiplexable assays, such as digital microfluidics and tandem mass spectrometry, were developed.”

- lines 20-22 “In particular, enzyme-based methods miss a large number of affected females.”

lines 20-24, join together concepts that should be separated out such as methods being developed and then addressing limitations of NBS such as enzyme testing on females to improve idea flow.

R: Thank you for the evaluable comment. We have rewritten the paragraph as follows: “Recently DNA-based methods have been applied to newborn screening in some countries. Using these methods, several newborn screening pilot studies and programs have been implemented worldwide. However, several concerns persist, and newborn screening for Fabry disease is still not universally accepted. In particular, enzyme-based methods miss a relevant number of affected females.”

Paper:

line 48-52, consider providing more detail on "childhood" vs. "youth" as there is stronger data available based on age categories.  "Youth" is too unclear a category to make sense in this context.   You have already referenced the Hopkin et al paper that can address much of that timing.  Consider Laney et al Genet Med, 17(5), 323-330. doi:10.1038/gim.2014.120  to better address childhood onset of symptoms to support NBS and need for early diagnosis.

R: We thank the reviewer for the important comment. We modified as follows: “Patients with a classic phenotype present with angiokeratomas, neuropathic pain, hypohidrosis, hearing loss, gastrointestinal symptoms. These symptoms can occur in early childhood before age 5 years, especially neuropathic pain and gastrointestinal symptoms [Laney et al, 2015]. In adulthood, the patients can show severe involvement of kidney, heart, central nervous (mainly cerebrovascular disease) and peripheral nervous system.”

In this section also consider reworking the separation out of females in the natural history discussion.   As females under age 5 with classic FD do have symptoms and in later onset do have cardiac/renal issues, consider just referring to classic and later onset FD without qualifying males/females and then the final statement can comment on phenotypic variability in females.

R: We thank the reviewer for the comment. We have eliminated the separation among males and females and modified the last sentence: “The clinical manifestations in female heterozygotes also depend on the X-chromosome random inactivation, that increases the phenotypic variability.” 

Also, the phenotype in women goes beyond X-inactivation, please add some nuance to that piece of natural history too.  (line 52, line 61, and other locations) See articles by Michael Beck  Beck M, Cox TM. Comment: Why are females with Fabry disease affected? Mol Genet Metab Rep. 2019 Oct 22;21:100529. doi: 10.1016/j.ymgmr.2019.100529. PMID: 31687338; PMCID: PMC6819736.and Esther Maier https://onlinelibrary.wiley.com/doi/10.1111/j.1651-2227.2006.tb02386.x

R: Thank you for the comment that allow us to improve our manuscript. We have added that “Unlike other X-linked disorders, females with Fabry disease often show clinical manifestations. One possible explanation, besides X-inactivation and skew deviation, is the ineffective cross-correction of the enzyme activity in vivo. Unaffected fibroblasts from Fabry heterozygotes mostly secrete the mature form of the enzyme, which lacks the high-uptake mannose-6-phosphate residues. This form cannot be efficiently endocytosed by the affected cells. Therefore, less active enzyme can complement the activity of the cells lacking expression of the enzyme [Beck et al, 2019].” 

Figure 2:  consider adding depression/anxiety too CNS manifestations

R: Thank you for the comment. We have modified the figure 2, including only cerebrovascular disease. 

Line 62, remove ( ) comment about X-inactivation as it doesn't add to discussion here and is oversimplified

R: Thank you for the suggestion. We have removed the sentence in brackets. 

In results section/screening methods: consider beginning the section with lines 89 explaining why different approaches are more frequently used in the beginning to set the stage for the comparison beginning line 115. Otherwise it is not clear why those 2 are being pulled out for comparison.

R: Thank you for the important comment. We have added that: “Among these methods, the most frequently used are digital microfluidics (DMF) and tandem mass spectrometry (MS/MS) because they are multiplexable with commercially available reagents.” 

Table 2:  rework table to provide consistent data in results section for easier visual comparison. Also comment on if the PPV includes VUSs or variants.

R: Thank you for the comment that allow us to improve the manuscript. To improve clarity, we have reported in the text the results showed in table 2. “These data have been confirmed by retrospective comparative studies in Taiwan and USA.”… “However, a prospective comparative effectiveness study on 89,508 deidentified residual newborn DBS performed in California demonstrated that MS/MS, DMF and immunocapture showed high sensitivity, but lack in specificity, with need to be improved.” 

Section 3.3.  It seems important to handle this DNA-based methods as a section and remove the direct correlation to females. The challenges to diagnosing females are very important but the paper would be clearer if the issues was separated from the DNA-based methods.  Perhaps a separate section.

R: Thank you for the valuable comment. We have removed the correlation between DNA-based methods and females. We discuss the challenges to diagnose females in the “BENEFITS AND CHALLENGES OF FD NEWBORN SCREENING” section. 

line 206, consider replacing "parent" advocacy with "patient/family advocacy" to better reflect issues

R: Thank you for the suggestion. We have changed “parent advocacy” with “patient/family advocacy”. 

Table 4- Given size of table, format to fit on one page or repeat headers to it can be effectively read. Also comment on confirmed patients from genetic analysis if it includes VUS/A143T and perhaps number of classic/later onset in confirmed column.  The incidence should be classic/later onset as well.

R: Thank you for the important observation. We have highlight that “confirmed patients” are patients carrying a GLA variant (including VUS and A143T). Unfortunately, many studies do not differentiate between classic and later-onset forms and do not report clinical follow up of the positive patients. Moreover, the classification of variants is changed over time as new data are available. For that reason, it is not possible to calculate a definitive incidence according to the specific phenotype. 

Have you considered combining data from same NBS testing programs if the publications are on the same population/methodology?

R: Thank you for the suggestion. We have combined data from some NBS programs in Taiwan (same laboratory and methodology). Studies in Italy and Japan were performed independently and carried in different areas of the country.  

Section 3.6.  Consider a short paragraph about the advantages of NBS for FD line 352 before jumping into challenges. Have you considered information on the early onset of symptom as an addition to support for NBS in FD?

R: We thank the reviewer for the important comment. We have added that “As above discussed, reliable and effective methods for screening on DBS are available. The disease is more prevalent than previously clinically estimated. Newborn screening allows the early diagnosis and treatment, leading to a better prognosis. Moreover, the diagnosis of a newborn can be followed by family counselling and screening.” 

Line 382.  I'm not sure this conclusion is fully supported yet.

R: Thank you for your comment. We have modified the text with “Therefore, the pathogenicity of this variant is still debatable”. 

Table 6: disadvantages- Diagnostic testing can distinguish infantile from non-classic in many cases.   Consider modifying this cell.

R: Thank you for the suggestion. We have modified the text with “Phenotype prediction can be difficult”.
